# A New Method Based on a Zero Gap Electrolysis Cell for Producing Bleach: Concept Validation

**DOI:** 10.3390/membranes12060602

**Published:** 2022-06-10

**Authors:** Rihab Belhadj Ammar, Takoua Ounissi, Lassaad Baklouti, Christian Larchet, Lasâad Dammak, Arthur Mofakhami, Emna Selmane Belhadj Hmida

**Affiliations:** 1CNRS, ICMPE, Université Paris-Est Créteil, UMR 7182, 2 rue Henri Dunant, 94320 Thiais, France; belhadj.rihab25@gmail.com (R.B.A.); larchet@u-pec.fr (C.L.); 2Laboratoire de Chimie Analytique et D’électrochimie, Département de Chimie, Faculté des Sciences de Tunis, Campus Universitaire, Tunis 2092, Tunisia; ounissi.takoua@gmail.com (T.O.); emnaselmane@gmail.com (E.S.B.H.); 3Department of Chemistry, College of Sciences and Arts at Ar Rass, Qassim University, Ar Rass 51921, Saudi Arabia; blkoty@qu.edu.sa; 4Gen-Hy, Rue de la Soie, 94310 Orly, France; amofakhami@gen-hy.com; 5Institut Préparatoire aux Études D’ingénieurs El Manar (IPEIEM), B.P 244, Tunis 2092, Tunisia

**Keywords:** bleach production, composite membrane, ion-exchange membrane, membrane electrolysis, zero-gap electrolysis cell

## Abstract

Commercial bleach (3.6 wt% active chlorine) is prepared by diluting highly concentrated industrial solutions of sodium hypochlorite (about 13 wt% active chlorine) obtained mainly by bubbling chlorine gas into dilute caustic soda. The chlorine and soda used are often obtained by electrolyzing a sodium chloride solution in two-compartment cells (chlorine-soda processes). On a smaller scale, small units used for swimming pool water treatment, for example, allow the production of low-concentration bleach (0.3 to 1 wt% active chlorine) by use of a direct electrolysis of sodium chloride brine. The oxidation and degradation reaction of hypochlorite ion (ClO^−^) at the anode is the major limiting element of this two-compartment process. In this study, we have developed a new process to obtain higher levels of active chlorine up to 3.6%, or 12° chlorometric degree. For this purpose, we tested a device consisting of a zero-gap electrolysis cell, with three compartments separated by a pair of membranes that can be porous or ion-exchange. The idea is to generate in the anode compartment hypochlorous acid (HClO) at high levels by continuously adjusting its pH to a value between 4.5 and 5.5. In the cathodic compartment, caustic soda is obtained, while the central compartment is supplied with brine. The hypochlorous acid solution is then neutralized with a concentrated solution of NaOH to obtain bleach. In this work, we studied several membrane couples that allowed us to optimize the operating conditions and to obtain bleach with contents close to 1.8 wt% of active chlorine. The results obtained according to the properties of the membranes, their durability, and the imposed electrochemical conditions were discussed.

## 1. Introduction

Sodium chloride electrolysis is a practical means of producing hypochlorite for the purpose of sterilization or disinfection procedures. It only requires salt and energy input and works under particular but well-controlled safety conditions in domestic or industrial installations.

This method is based on the generation of chlorine by anodic reduction of chloride. There are many variants of the process that allow very different forms and concentrations to be generated, ranging from diluted solutions that can be directly injected into drinking water or disinfection circuits to industrial-scale production of concentrated chlorine and soda. The process can work with salt solution but it can also use brackish water or even seawater. In all cases, a purification phase will be necessary to avoid fouling problems due to precipitation of insoluble salts.

Many elements can vary between the different installations; including the presence or absence of a separator, the nature of the separator, the nature of the electrodes, and, finally, the several possibilities of electrical connection—in series or in parallel—and of hydraulic connection—with or without recycling the products. Whatever the variant of the process used, the choice of electrodes and membranes that can be used is greatly reduced in order to have a minimum longevity of several years due to the oxidizing or basic nature of the reaction products. The nature of the materials used for the anode and cathode also plays an important role in the energy balance due to the voltage surge required for the reactions envisaged. In each case, operating parameters such as salt concentration, solution flow rate, cell dimensions and geometry, and voltage/current parameters must be carefully optimized. As chlorine and caustic soda are important raw materials and are mass-produced, many recommendations have been published by the European Commission [1] and Smith [2] concerning the process choice, and much research is still ongoing to improve yields and energy costs.

### 1.1. Electrolysis Cells

#### 1.1.1. Reactions

The expected reaction at the anode is the oxidation of chloride to chlorine, but due to the oxidation potentials, water will also oxidize to form hydrogen. At the cathode there will be reduction of water to form OH^−^ ions and hydrogen. Depending on the different parameters of the cell constitution or its operation, other parasitic side reactions may also occur, such as chlorate formation [3,4].
12HClO + 6H_2_O → 4ClO_3_^−^ + 8Cl^−^ + 24H^+^ + 3O_2_ + 12e^−^
2HClO + ClO^−^ → ClO_3_^−^ + 2Cl^−^ + 2H^+^

These side reactions are repressed by lowering the pH value [5].


**Reaction Step I:**
Anode:    2H_2_O → 4H^+^ + O_2_ + 4e^−^      (I-a1)
     2Cl^−^ → Cl_2_ + 2e^−^         (I-a2)
 Cathode:      2H_2_O + 2e^−^ → H_2_ + 2OH^−^     (I-c1)



**Reaction Step II:**
2Na^+^ + 2OH^−^ + Cl_2_ → NaOCl + H_2_O + Na^+^ + Cl^−^ (II-1)

 H_2_O + Cl_2_ → HOCl + H^+^ + Cl^−^         (II-2)


At the anode, chloride ions are converted into gaseous chlorine:2Cl^−^ ⇆ Cl_2_ + 2e^−^    E^0^(Cl_2_/Cl^−^) = 1.358 V

The cathode compartment is fed with a water solution of sodium hydroxide, i.e., pH of 14. At the cathode, water is reduced to gaseous hydrogen and hydroxyl ions:2H_2_O + 2e^−^ ⇆ H_2_ + 2OH^−^   E^0^ (H_2_O/H_2_) = 0.828 V

The hydrogen gas produced at the cathode together with caustic solution (concentration 35 wt%) from chlor-alkali cells is normally used for the production of hydrochloric acid or as a fuel to produce steam and energy [6].

For disinfection purposes, hypochlorous acid is produced by the reaction of chlorine which is disproportionate in the presence of water Cl_2_ + H_2_O ⇆ HClO + H^+^ + Cl^−^.

#### 1.1.2. Without Separator

It is theoretically possible to prepare hypochlorite directly by making electrolysis of a NaCl solution, leaving the reaction products in contact at the anode and cathode according to reaction II-1.

Various side reactions may also occur, depending, in particular, on the electrodes used and the operating voltage, such as the oxidation of hypochlorite to chlorate [4,5,7].

Although this type of cell has the advantage of simplicity, it has many disadvantages. This includes the low concentrations obtained (<1% active chlorine), the impossibility of separating the production of chlorine and soda if desired, and the release of a dangerous mixture of hydrogen and oxygen. 

#### 1.1.3. With Separator

The introduction of a separator makes it possible to separate the production of chlorine and soda and to avoid the hydrogen oxygen mixture. However, it is necessary that it allows electrical charges to pass through by ionic transfer. To this end, two solutions are used: either a simple porous system (diaphragm) or a selective system for ionic transport. In all cases, they must introduce the lowest possible resistance so as not to penalize energy consumption. The materials used must also be able to resist the oxidizing species produced.



*Diaphragm*



The porous separators were initially made of asbestos. With the prohibition of the use of this material, different solutions were proposed. The most common solution is to use a porous Teflon, sometimes supplemented with inclusions of particles such as ceramics or ion-exchangers [7]. An alternative solution entails the use of conductive ceramic membranes [5].



*Ion-exchange membranes (IEMs)*



Ion-exchange membranes are polymeric membranes containing ionizable functional sites of positive (AEM) or negative (CEM) charge. They are appealing thanks to their good electrical conductivity and their transfer selectivity. However, they must be resistant to oxidizing environments. Therefore, perfluorosulfonic polymers are preferred in case a separate production of chlorine and sodium hydroxide is desired. The addition of a layer bearing carboxylic functions makes it possible to practically avoid any transfer of OH ions and leads to soda yields of up to 50% [5,8,9]. 

Due to the high cost of this type of material, other types of membrane have been the subject of research. In 2021, Kim et al. [5] obtained similar productions with Nafion^®^117 and 324 industrial membranes as with homemade SPEEK membranes for the production of low-concentration hypochlorite applicable to water treatment which does not require only concentrations below 1000 ppm. The three membranes give equivalent yields and concentrations. The use of SPEEK membranes, therefore, makes it possible to reduce costs thanks to the lower price of the membranes but also thanks to their lower electrical consumption. However, no lifespan study has been carried out. 

In 2020, Mohammadi et al. [10] compared the use of an anionic membrane based on polystyrene/divinyl benzene, including quaternary ammonium functions with two cationic membranes having sulfonic functions, with or without a Teflon weft. Operating parameters are selected based on use by direct injection into water supply systems with chlorine concentrations ranging from 355 to 916 mg·L^−1^. They show that the best quality of the product preservation is obtained with the Teflon reinforced cationic membrane; however, the quality of production with MEA proves to be mediocre. In the same study, Mohammadi et al. [10] also use a bipolar membrane whose performance is intermediate compared to the other two types. This type of process with MEI separator is reputed to be the most economically appealing. It tends to currently replace the old devices.

### 1.2. Electrodes

Regardless of the process type, the anode is in contact with very reactive oxidants and must therefore meet specific constraints. The first electrodes used were made of graphite, then electrodes, made of platinum, diamond, etc., were developed [9]. Currently, the most common solution is the use of dimensionally stable anodes (DSA) consisting, for example, of titanium covered with a deposit of metal oxides such as ruthenium, iridium, or titanium [4,9].

The cathode in contact with OH^−^ ions poses less of a problem from the point of view of corrosion. While stainless steel electrodes may be suitable, energy criteria lead to the adoption of more sophisticated materials. This is because the choice of electrodes also plays on the optimization of the current/voltage parameters by their influence on the overvoltages necessary for the activation of the reactions and therefore on the cost of energy operation. For this reason, activated nickel-based cathodes coated with a catalyst including nickel and platinum group elements are used [9,11]. 

### 1.3. Cell Design

To achieve important reduction of the cell resistance, it is necessary to significantly reduce the thickness of electrolytes. The first idea is to reduce the space between electrodes while avoiding a gas blockage between the electrode and the membrane. The optimal thickness is between 0.2 and 1 mm [10]. The distance between the cathode and the membrane is typically set at approx. 1 mm.

Another solution is to change the position of the different elements. In the simplest case, porous electrodes are placed directly in contact with the membrane, hence the name zero-gap [6]. A more complex setup can also be used. In this case, the membrane is placed in contact with the cathode by means of a mesh glued to it and connected to a cathode by means of an elastic metal element, thus ensuring the electrical conductivity between the two elements of the cathode [12,13].

### 1.4. Monopolar or Bipolar Electrolyzer

In the case of a system using several unit cells, it is possible to make two types of electrical connections. These can be powered in parallel (unipolar connection with high current circuit and low voltage) or in series (bipolar connection with a low current and high voltage). 

Bipolar electrolyzers are preferred due to the reduced investment (simple filter press design leading to easier manufacturing) and operating costs (better energy performance due to smaller voltage drop) as well as to easier maintenance (easier detection of faulty cells by monitoring individual cell voltages, shorter duration of shutdown and start-up phases to replace membranes) [3,14,15].

### 1.5. Recycling

In the case of the production of a low-concentration disinfectant, it is necessary to optimize the transformation of the produced chlorine. Kim et al. [5] propose reinjecting the anodic chlorine into the cathodic solution containing the formed soda. Production is then significantly improved, whereas an assembly without recirculation yields a production practically equivalent to an assembly without a separator.

### 1.6. Parameters’ Optimization

The chlorine-soda process operates industrially at temperatures above 60 °C, most often around 90 °C, and the cell is supplied with a saturated NaCl solution. For the formation of disinfectant solutions, it is easier to work at room temperature. On the other hand, due to the disproportionation of chlorine being an exothermic reaction, it is sometimes necessary to provide a method to cool the reaction medium. The activity and the quality change during storage time also depends on the operating parameters. 

The solution used is more dilute to avoid too much salt transport but must remain sufficiently concentrated to limit electric resistance. The concentration of the electrolyte at the anode can vary from 180–240 g·L^−1^ [5,16] to 1.65–3.5 g·L^−1^ at pH between 1.0 and 4.5 [10]. Liao et al. [17] studied the influence of the flow rate, NaCl concentration, electrolysis time, and the current density on the solution activity. All obtained curves have a similar shape with an optimal result for a short electrolysis time of 10 s. Although the variation of the parameters is small, the best quality is visibly obtained for the lowest flow rate (10 mL·s^−1^), the highest salt concentration (0.2 g·L^−1^), and the highest current density (0.4 A·cm^−2^). 

### 1.7. Salt Quality

When the salt solution is not obtained by redissolving purified NaCl but from recycled solutions, or solutions prepared with non-pure salt, brine, or seawater, the precipitation of bivalent salts will obstruct the membrane and, hence, its resistance. The presence of transition metals can have a negative catalytic effect which leads to the appearance of undesirable secondary products [1,18].

### 1.8. Other Cells

Another means of reducing energy consumption can be obtained by introducing oxygen-depolarized cathodes (ODC). This method consists of modifying the reaction on the cathode by injecting oxygen by means of a gas diffusion electrode (GDE); a technique well known from alkaline fuel cells. By changing the reaction of hydrogen production by a reduction in oxygen, up to 30% energy can be gained [9,19,20,21,22].

A very different approach is proposed by Hou et al. [23], who return to a cell without a separator but based on a different operating principle from conventional electrolysis. The process uses a pair of electrodes based on Na_0.44_MnO_2_ operating on a two-phase cycle; the first, in NaOH 1 M, produces the cathode H_2_ and OH^−^ from water and deintercalates the anode in Na^+^, the second, in saturated NaCl medium, inserts Na^+^ in the cathode and produces chlorine on the anode. The process leads to a Faradic efficiency of 100% for hydrogen but it is only 90.2% for chlorine against 97.4% for electrodialysis.

For the direct disinfection of water, Isidro et al. [19] compared the use of two electrolysis cells with boron-doped diamond (BDD) coatings but varying by geometry and the flow conditions. One is a flow-through cell with perforated electrodes; the other is a zero-gap cell in which a Nafion membrane separates the anode and cathode. The performances are quite identical, but the second one minimizes the production of chlorates and perchlorates when operating in a single-pass mode.

The main purpose of this study is to develop a new method based on a zero-gap electrolysis cell for producing bleach at a relatively high concentration. To generate hypochlorite ions in a more efficient, controllable, and cost-effective way, we have not only used different types of commercial membranes such as ion-exchange membranes (Nafion^®^, AMX, CMX, Fuji AEM) and a composite membrane (Zirfon^®^, AGFA, Mortsel, Belgium), but also a homemade BN/PTFE composite membrane, presenting high chemical resistance. This study offers a new possibility to develop an efficient and stable hypochlorite ion generation system in a batch mode.

## 2. Materials and Methods

### 2.1. Materials

We used the Zirfon membrane (AGFA, Mortsel, Belgium) which is one of the commercially available low-cost membranes with very low electrical resistance. This membrane will be used to compare the performance of our BN/PTFE membrane, which was synthesized by laminating a paste based on boron nitride (BN) and polytetrafluoroethylene (PTFE). PTFE provides chemical resistance to the membrane and BN provides thermal resistance and porosity.

Zirfon and BN/PTFE membranes are porous and are likely to have significant gas and/or ion leakage. We therefore considered dense perfluorinated cation-exchange membranes; in this case, Nafion^®^117 and Nafion^®^911 were used. This type of perfluorinated membranes is replacing separators and diaphragms in aging installations. However, their rather high price hinders their deployment. We have, therefore, chosen two other dense membranes to see their behavior in this type of cell: CMX (Astom, Tokyo, Japan) and AEM type I (Fuji, Tilburg, The Netherlands).

For electrochemical experiments, high-purity pellets of sodium chloride (99.99%; AXAL Pro, Paris, France) and 6 M sodium hydroxide solution (VWR, Paris, France) were used.

### 2.2. Characterization of the BN/PTFE Prepared Membrane

Thermal properties, particularly thermal stability of the produced membrane, were operated by thermogravimetric analysis through an analysis of thermogravimetric data (TGA) (LABSYS evo TGA 1150, Lyon, France). The samples were heated at a scanning rate of 10 °C/min from room temperature to 750 °C in argon atmosphere. Each sample had 5–10 mg.

The mechanical properties of the membranes were measured by a bicolumn traction machine (INSTRON 5965, Norwood, MA, USA) and a microcomputer-controlled electronic universal testing machine (MTS) with a test speed of 1 mm/min at room temperature. The sample membrane was cut into small splines of standard size, and the digital calipers instrument was used to measure the average thickness of the samples. The tensile strength and elongation at break were finally obtained through stress–strain curves.

The morphology study is important to confirm the homogeneity of polymer blends. The surface morphology of the BN/PTFE samples was investigated by scanning electron microscopy (MERLIN scanning electron microscope by ZEISS associated with a GEMINI II column, Léna, Germany). Before SEM measurements, all of the samples were coated with a thin Pt/Pd layer.

Conductivity K_m_ of the membrane is an intrinsic characteristic for membrane separation. It was measured using the standard method as detailed in [24], based on the use of a clip-cell and a thermo-regulated water bath at 25 ± 0.5 °C and a CDM92 conductivity meter (Radiometer-Tacussel) operating at 1 kHz AC.

The water content (*W_c_*) measurements for some membranes were obtained by immersing a sample for 24 h at 25 °C in deionized water. The hydrated mass (*W_h_*) is quickly measured after removing or wiping out any remaining surface water with a paper. The dry mass (*W_d_*) is obtained after a drying process at 80 °C until the membrane weight becomes stable using a moisture analyzer HB43-S Mettler-Toledo. The *W_C_* (%) was found as follows:$$W_{c}=\frac{W_{h}-W_{d}}{W_{h}}\times 100$$

### 2.3. Cell and Device Description

Hypochlorites (HOCl and NaOCl) are generated electrochemically using the zero-gap electrolysis cell described in Figure 1, where we distinguish the three compartments; each one is delimited by a seal enclosing a grate. The separation between two compartments is ensured by the membranes (ion-exchange or composite) and metal filters ensuring the homogeneity of the solution flow and the evacuation of the formed gases. The electrodes are placed on both sides of the sandwich, held in contact with the membranes and electrically isolated from the two stainless steel blocks by EPDM pads. The tightness is ensured by a moderate tightening of the whole with eight threaded rods. For simplicity’s sake, the operating principle of this cell is presented in Figure 2. Generally in industrial chlor-alkali advanced processes, the anodes used in membrane cells are commonly made of titanium (Ti) coated with a mixture of different metal oxides, and the cathodes are made of steel or nickel alloy [25]. According to the research context, titanium with a platinum coating Ti/Pt anode and a stainless steel cathode were employed in the electrochemical cell to yield the highest hypochlorite ion production efficiency during the electrolysis process.

In the zero-gap electrolysis cell, the interelectrode distance is determined by the thickness of the membrane. It consists of a mini-stack of three elementary cells separated by two membranes, a feed compartment that circulates the brine water, and anodic and cathodic chambers where electrochemical reactions are implemented.

The various manipulations were performed under a constant voltage and a flow rate in the electrolysis cell, maintaining them at room temperature to prevent excessive oxidation of hypochlorite to chlorate according to the following reaction [15]:3ClO^−^⇆2Cl^−^+ClO_3_^−^

The concentration of hypochlorite ions in the product solution was measured by titration with 0.1 M sodium thiosulfate (Na_2_S_2_O_3_) solution, using the mixture of potassium iodide, hydrochloric acid, and aqueous starch solution as an indicator. The following equations were applied to calculate the concentration of hypochlorite ions and the chlorometric degree of the sample according to the standard method [26]:$$\left[{\mathrm{ClO}}^{-}\right]=\frac{C_{2}\times V_{2}}{2\;V_{1}}$$
$$Chlorometric\;degree\;\left({}^{\circ}Ch\right)=\left[{\mathrm{ClO}}^{-}\right]\times 22.4$$

$C_{2}$: Sodium thiosulfate concentration.

$V_{1}$: Sample volume.

$V_{2}$: Sodium thiosulfate volume.

## 3. Results and Discussion

### 3.1. The Operating Mechanism of the Hypochlorite Zero-Gap Electrolysis Cell

For the electrochemical production of hypochlorous acid and sodium hydroxide, the NaCl solution serves as both an electrolyte and a reagent. The detailed schematic description of the electrolysis cell and the mechanism of the electrochemical redox reaction generating HOCl and NaOH are shown in Figure 2, according to the following chemical reactions.

The proposed mechanism for the generation of hypochlorite ions involves the following set of redox reactions occurring at the electrodes [23]:

At the anode: Cl^−^ → ½Cl_2_ + e^−^(1)
Cl_2_ + H_2_O → HClO + H^+^+Cl^−^(2)
2 H_2_O → O_2_ + 4 H^+^ + 4 e^−^(3)

At the cathode:H_2_O + e^−^ → H_2_ + OH^−^(4)

Inside the chambers, Cl^−^ and sodium (Na^+^) ions migrated toward the attracting anode and cathode, respectively [23]. As shown in the equations, water (H_2_O) and Cl^−^ are oxidized at the anode to produce oxygen (O_2_) (Equation (3)), hydronium ions (H^+^) (Equation (2)), and molecular chlorine (Cl_2_) (Equation (1)). On the surface of the anodic electrode, Cl_2_ gas molecules react with adjacent water molecules and produce hypochlorous acid molecules (Equation (2)).

Based on preliminary tests, it can be noted that the combination of two NaOH and HClO outputs produces a NaClO mixture with a chlorometric degree lower than 0.5. This can be explained by the fact that the mixture remains basic, which is unfavorable to the formation of HClO according to the stability diagram of hypochlorites presented in Figure 3.

The pH of the solution defines the proportions of the different forms of chlorine elements: Cl_2_, HClO, and ClO^−^. Since in the anode compartment there is a formation of H^+^ ions, the pH of this compartment will decrease, making the Cl_2_ form predominant (see Figure 3). However, our method is based on the exclusive formation of the HClO form in the anode compartment. We have, therefore, readjusted the pH of this compartment by using the system of measurement and adding a 5 N NaOH solution introduced in Figure 2 (often a few drops fall every minute). We purposely chose a very concentrated NaOH solution to limit the dilution effects of the anode solution. As shown in Figure 3, the pH of the anode compartment is maintained between 4.5 and 5.5.

### 3.2. Electrolysis Voltage Determination

Below a certain voltage threshold, no electrolysis appears to be thermodynamically possible. Therefore, it is very important to determine the minimum voltage that we must apply to make the electrolysis possible in our case. In addition, this parameter depends on several factors, including concentration and nature of the electrolyte, ionic strength of the solution, nature of the electrodes, working temperature, pH solution, etc. Therefore, we used the same cell previously described in Figure 3 in its two-compartment configuration (eliminating one of the two membranes), and determined the intensity–potential curves using a saturated NaCl solution in both compartments. We have plotted in Figure 4 two examples of curves, one for the commercial Zirfon^®^ membrane and the other for our BN/PTFE membrane.

We notice that the curves are similar, even parallel, between the two membranes. Generally, we have a slow variation similar to an oxidation wave for potentials lower than 1.3 V, followed by a very fast variation, reminding us of a water oxidation wall. The voltage at the transition from one regime to another is determined by the tangent method. We obtain 1.2 V with BN/PTFE and 1.3 V with Zirfon membrane. We also remark that the I–E curve of the BN/PTFE membrane is always above the curve of the Zirfon membrane, which means that for the same voltage value applied to the system, there is more current flowing through the BN/PTFE than through the Zirfon. Thus, higher rates of oxidation–reduction reactions for the BN/PTFE are observed.

Zhu et al. [28] studied the electrochemical behavior of their membranes by plotting the I–E curves for 0.05 M NaCl solution and Ti/RuO_2_ anode in the presence or absence of other solutes. The obtained results show that the curve shape is identical to ours except that the current densities are significantly lower than what we obtain while the angular point potential, ~1.2 V, is very close to what we obtained. 

Lim et al. [29] have also established the CER performance curves of Pt/CNT, PtNP/CNT, DSA (dimensionally stable anode), and CNT (carbon nanotubes) catalysts in 0.1 M HClO_4_ + 1.0 M NaCl at an electrode rotation speed of 1600 rpm. They obtained a curve very similar to ours and showed that the nature of the catalytic deposit significantly influences the angular potential point. The best curve is obtained for the Pt/CNT deposit.

All these elements show that our cell, with its electrodes and the considered membranes, allows to have electrochemical conditions rather favorable to carry out the operations of oxidation of Cl^−^ to have mainly Cl_2_ and ClO^−^, and water reduction to have mainly OH^−^ ions. In the rest of our study, we will use potentials higher than 1.3 V but lower than 6 V because experience shows that at this potential value the electrodes start to degrade.

### 3.3. Main Proprieties of BN/PTFE and IEMs Used Membranes 

#### 3.3.1. Morphology

Figure 5 shows SEM images of the surface and cross-section of the BN/PTFE membrane. It appears that our composite membrane has a symmetrical morphology with a thickness of 400 ± 10 µm as revealed by the cross-section image. The surface image shows that ceramic particles are homogeneously distributed on the sample surface and they give the membrane a slightly rough appearance. These particles, in the majority of cases, are held together by a network of PTFE filaments, providing both good mechanical strength and a high percolation threshold. The cross-section of the membrane confirms these last two points.

Furthermore, several studies have shown that the ceramic/PTFE blend provides a porous structure composed of nodes connected by fibrils as revealed in the SEM images of our membrane. Similarly, Bousbih et al. [30] reported that the formation of a porous structure is due to the penetration of PTFE polymer between the ceramic layers.

#### 3.3.2. Thermal Stability

The thermal properties of boron nitride and BN/PTFE membrane have been investigated using a thermogravimetric analysis (TGA) in argon atmosphere. The following thermograms are presented in Figure 6. The TGA curve of BN ceramic material shows that there is no change in its weight, which confirms its exceptional thermal stability. However, the TGA and DTG curves for the membrane show three main waves, indicating successive degradation. Weight loss between room temperature and 120 °C is easily attributed to the residual solvent (water and ethanol) evaporation. The temperature at which polymer degradation started (T_onset_) and the maximum rate degradation temperature (T_max_) of this crucial step were about 333 °C and 540 °C, respectively. Weight loss at these temperatures might be due to the degradation of the PTFE polymer, starting with the short PTFE chains. Pan et al. [31] confirmed that pure PTFE degradation occurs between 500 °C and 600 °C. They reported that filler insertion does not alter the thermal degradation mechanism of the PTFE matrix. Thus, we can conclude that our composite membrane exhibits a high thermal stability since the greatest mass loss occurs only at high temperatures compared to non-fluorinated polymer-based membranes.

#### 3.3.3. Mechanical Properties

The stress–strain curve of composite membranes is obtained by applying a tensile force at a uniform rate and constant temperature. The profiles of the stress–strain curve are strongly influenced by the type of polymer, the filler used, and their ratio to the matrix. The curve gives information on Young’s modulus, yield strength, breaking point, and elongation at break. The corresponding Young values and tensile strength to break ratio are illustrated in Figure 7. From the results, we can see that the tensile strength of our BN/PTFE membrane is 0.053 MPa with a Young’s modulus of 3.68 MPa and elongation at break of 110%. However, the deformation of the composite membrane is still lower than that of the pure PTFE film, as reported by Cao et al. [32]. This phenomenon might be explained by the fact that the bonding effect of BN to PTFE was weaker than the effect of BN adding on the decrease of intermolecular forces in PTFE. Moreover, comparing our membrane with the commercial Nafion^®^117 membrane, as shown in Figure 7, we remark that the elongation at break of Nafion^®^117 is high compared to our membrane. This can be attributed to the low amount of PTFE polymer introduced in the composite membrane BN/PTFE. Mechanically, our membrane is quite malleable and does not create strong stresses in the cell due to dimensional variations caused by swelling phenomena.

These obtained results are in agreement with the study developed by Huang et al. [33] who showed that the prepared BN-SPEEK/PTFE reinforced membranes show increasing resistance with the amount of BN. This effect was due to the fact that BN has a solitary electron pair and can produce “acid–base” pairs with the sulfonic acid groups existing in the membrane.

These results confirm that the composite membrane has good mechanical stability, which is mainly due to the high degree of fibrillation. This in turn promotes the formation of a plastic filler/polymer network and increases the flexibility of our membrane. 

### 3.4. Performance of the Hypochlorite Generation Process Using Different Membranes

Figure 8 shows the kinetic study and the performance of the hypochlorite generation process performed on different types of membranes. Observing the results obtained in Figure 8 and comparing the two composite membranes Zirfon^®^ and our BN/PTFE membrane in terms of efficiency in the production of hypochlorites, we note that under the same operating conditions and with the BN/PTFE membranes couple, we produced about seven times the amount produced by the Zirfon^®^ membrane couple. These results can be attributed to the ionic conductivity and to the chemical stability of the BN/PTFE membrane, as well as to the capacity of the BN charge for proton exchange, which favors the transfer and diffusion of the ions to the two compartments. Hu et al. [34] revealed that the excellent proton conductivity of h-BN at room temperature is higher than that of graphene at higher temperatures. The proton conductivity of h-BN arises from its polarized covalent bonding of the B and N atoms as a result of the difference in electronegativity and, therefore, causes valence electrons to accumulate around the N atom and form an uneven electron cloud distribution.

In addition to ion-exchange membranes, the production of hypochlorite ions varies according to the physicochemical characteristics of the membranes, as indicated in Table 1. Among the used membranes, the CMX/AMX pair led to better results compared to the other IEMs. This can be explained by their selectivity, good exchange capacity, and their high ionic conductivity, as well as their water content, which facilitates and promotes the ionic transfer through the membranes.

It should be noted that even after 8 h of operation, we did not observe any noticeable exothermic effect in the three solutions (+5 °C at most). Open-air cooling appears to be sufficient to maintain a temperature close to room temperature.

By using the BN/PTFE composite membrane on both anodic and cathodic sides at a current density of 16.6 mA.cm^−2^, the hypochlorite generation cell has attained its maximum production. It has yielded about 260 mM of HClO after 8 h of operation. Indeed, per unit of time, the amount of retro-diffusion becomes more important than the amount produced by the current. Due to this, we could not attain more than this efficiency that can be attributed to the porosity of the membrane, which has been already demonstrated and confirmed by SEM images (Figure 5).

The phenomenon of retro-diffusion was confirmed by the curves in Figure 9 by determining the amount of hypochlorite ions in the NaCl feed compartment through the use of three different pairs of membranes.

This effect is remarkable when using porous composite membranes. In addition, with our BN/PTFE membrane, the sum of the amount of hypochlorite ions produced at the anode and the part diffused in the feed chamber after 8 h of operation led us to obtain 11.4° chlorometric degree at room temperature with a current density of J = 16.6 mA.cm^−2^.

Besides the physicochemical properties of the used membranes, the difference in hypochlorite ion yield by zero-gap electrolysis may be related to the chemical stability of the membranes used and their resistance to aggressive media during operation. Table 2 allows to evaluate the chemical resistance of the employed membranes. It shows the states of the different membranes after 8 h of operation.

Generally, oxidation degrades the functional groups and some organic material. According to Gaudichet-Maurin et al. [44], exposure to NaClO produces the breakage of the macromolecular chain of polysulfone. Similarly, Prulho et al. [45] estimate that the immersion of the PES/PVP mixtures in a hypochlorite solution at pH = 8 causes the oxidation of the aromatic rings into phenol groups. This enables us to explain the degradation of Zirfon^®^ which contains 15 wt% of polysulfone [13]; after two tests, even though it has good physicochemical properties, its chemical stability and resistance to aggressive agent remains limited. In addition, the degradation of the surface of the Zirfon^®^ membrane, which is located against the central compartment of NaCl, is due to the presence of a quantity of hypochlorite ions in this compartment as a result of the retro-diffusion phenomenon explained previously.

The degradation of the IEMs-based poly(styrene-co-divinylbenzene) AMX and CMX can be attributed to the breakage of the polymer chain and the anionic and cationic functional sites. Garcia et al. [46] showed that sodium hypochlorite provoked a degradation of the quaternary ammonium sites of anion-exchange membranes and chain scission of the poly(styrene-co-divinylbenzene) backbone from anion and cation-exchange membranes through chain radical oxidation.

During operation and under the effect of the electric field, Na^+^ ions selectively migrate towards the cathode through a cation-exchange membrane. Moreover, due to the formation of hydroxide ions, OH^−^, following the electrochemical reaction which occurs at the cathode according to Equation (4), an important quantity of NaOH is produced during the operation. The results of the dosage are illustrated in Figure 10. According to this figure, the amount of NaOH produced depends on several factors such as the type of membranes and whether they are exchange or composite membranes. In this comparative study, one can reach about 2.5 M NaOH after 8 h using the CMX/AEM membrane pair.

The quantities of NaOH produced in the cathodic compartment differ from one pair to another, and this can be explained by the physicochemical characteristics of the used membranes. More specifically, the difference in exchange capacity, as well as the selectivity and the affinity, of the functional groups of the membranes to Na^+^ ions are observed.

## 4. Conclusions

The production of bleach at high concentrations by electrolyzing a saturated sodium chloride solution is not simple. Several parameters and factors are involved in both chloride oxidation and water reduction. In addition, the aggressiveness of hypochlorite ions and chlorine molecules is so important that few materials can resist them. 

In this study, different membranes were used. Some of them are commercial ion-exchange or porous composite membranes. In addition, a porous composite membrane based on boron nitride and Teflon was developed by us. 

We have shown that it is possible to reach active chlorine contents close to 1.8 wt%, which is significantly higher than the contents obtained by simple electrolysis with or without a separator (0.3 to 1.0 wt%) and that the limiting factor is the back-diffusion of the HClO produced in the anode compartment into the other compartments through the pores of the ceramic membranes. Only the Nafion^®^117 type membranes and our BN/PTFE membranes are chemically resistant, even after many operations.

This study allowed us to validate the concept of such a process, and we are currently working on the means to reduce the porosity of the BN/PTFE membranes in order to limit the loss of HClO species by back-diffusion in the other compartments.

## Figures and Tables

**Figure 1 membranes-12-00602-f001:**
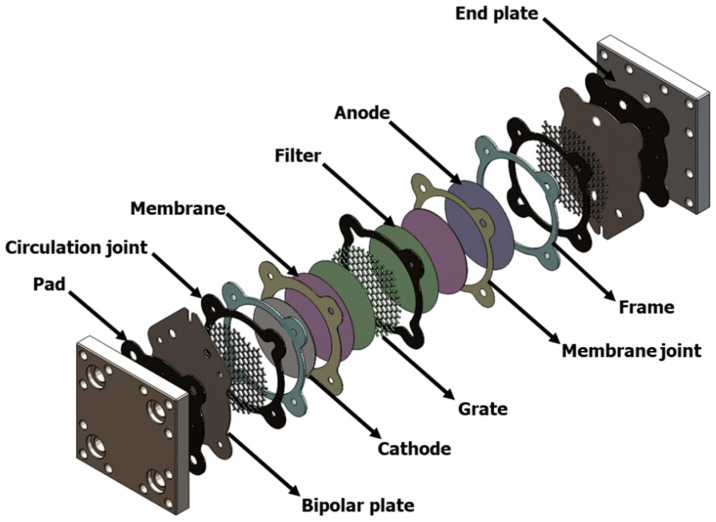
3D view of the electrolysis cell.

**Figure 2 membranes-12-00602-f002:**
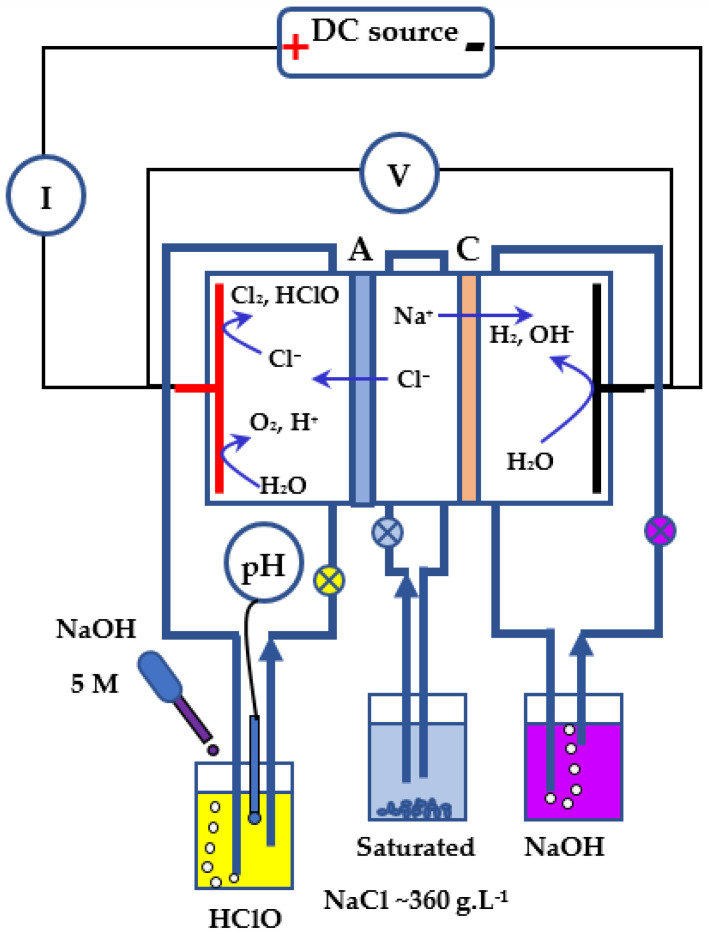
Experimental device with its measurement instruments and solutions circulations, and pH control system of the HClO compartment.

**Figure 3 membranes-12-00602-f003:**
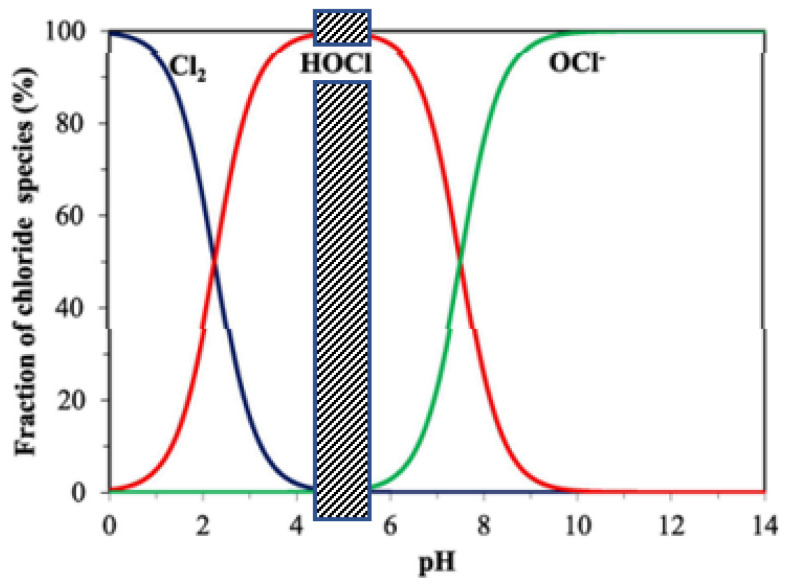
Fraction of aqueous active chlorine species as function of pH, calculated using a composition of 0.05 mol·L^−1^ NaCl at standard temperature and pressure (Adapted from [27]).

**Figure 4 membranes-12-00602-f004:**
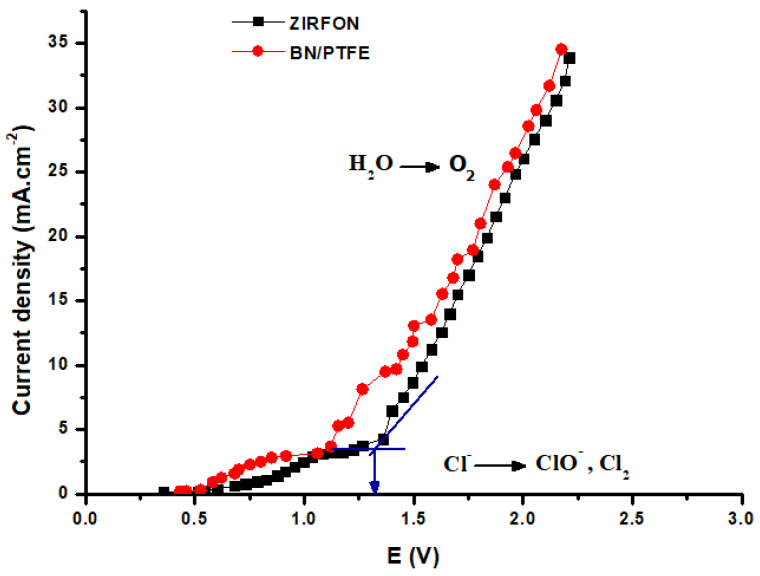
Current–voltage curves for Zirfon^®^ and BN/PTFE membranes obtained with the used cell and saturated NaCl solution.

**Figure 5 membranes-12-00602-f005:**
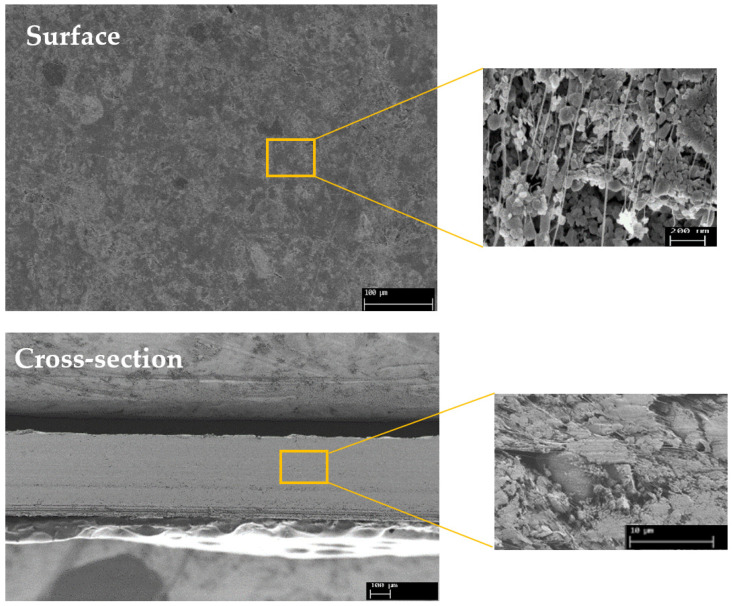
SEM images of the surface and cross-section of our prepared BN/PTFE membrane.

**Figure 6 membranes-12-00602-f006:**
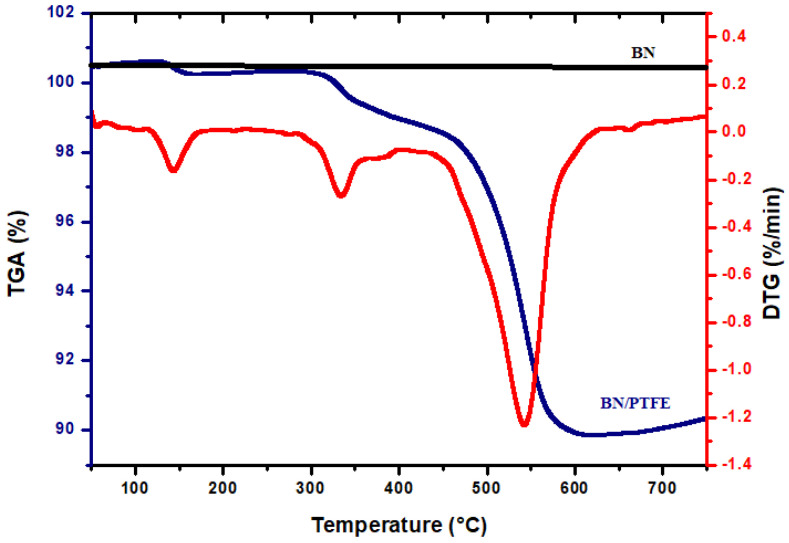
TGA analysis of boron nitride (BN) and BN/PTFE membrane.

**Figure 7 membranes-12-00602-f007:**
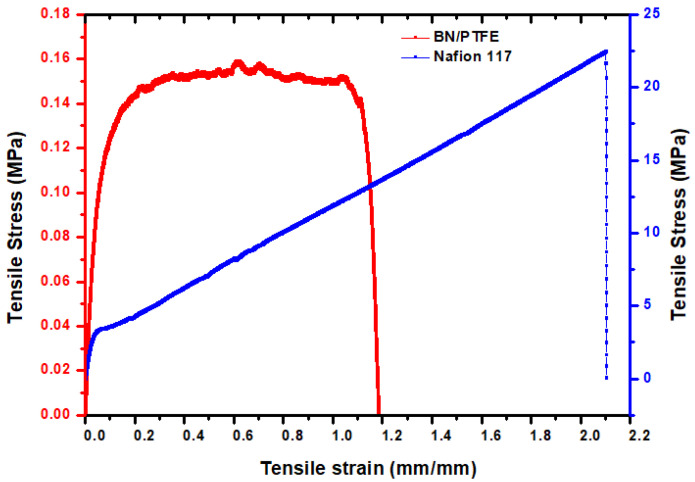
Representative stress–strain curve of BN/PTFE and Nafion^®^117 membranes at room temperature.

**Figure 8 membranes-12-00602-f008:**
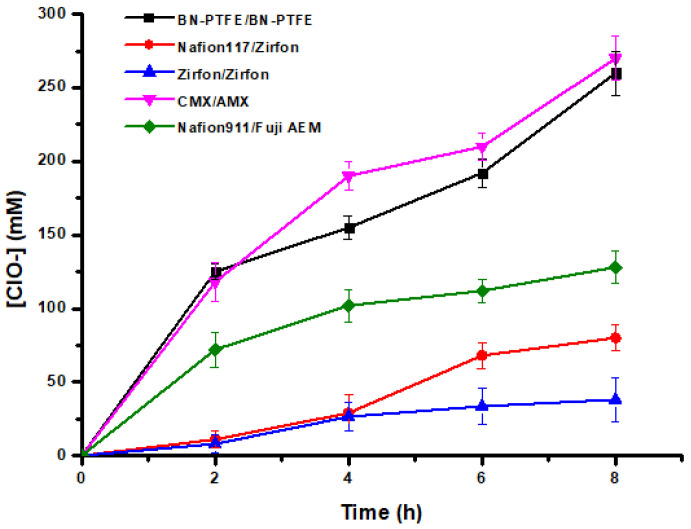
Kinetic curves of hypochlorite ion generation by electrolysis of brine using various types of membranes at (constant rate, room temperature, J = 16.6 mA.cm^−2^, 4.5 < pH_Anodic_ < 5.5).

**Figure 9 membranes-12-00602-f009:**
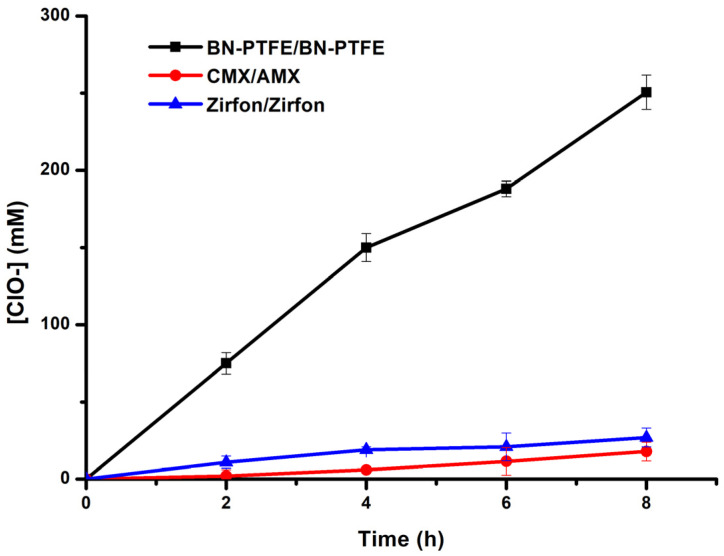
Back-diffusion curves: comparative study of the concentration of hypochlorite ions in the central NaCl compartment using different types of membranes under the same operating conditions (constant flux, room temperature, J = 16.6 mA.cm^−2^, 4.5 < pH_Anodic_ < 5.5).

**Figure 10 membranes-12-00602-f010:**
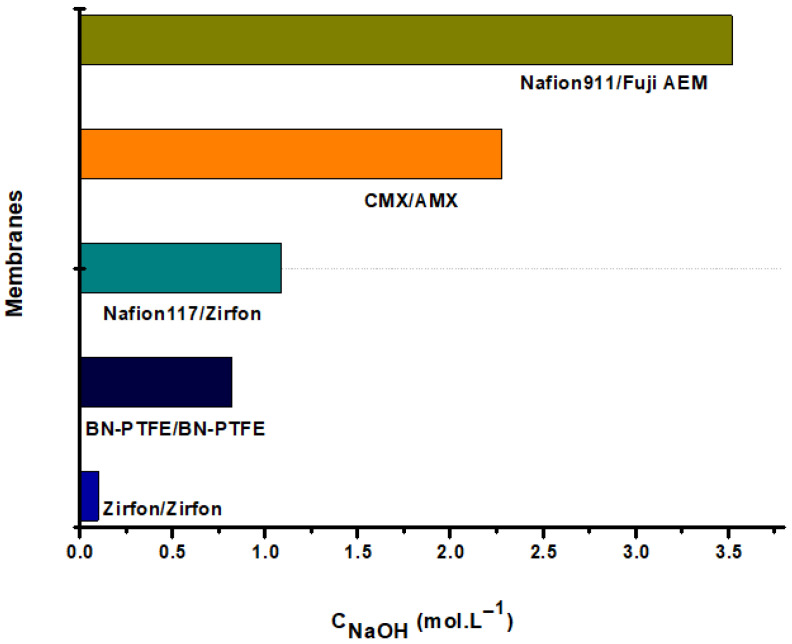
Quantity of NaOH generated in the cathodic compartment with I = 500 mA.

**Table 1 membranes-12-00602-t001:** Main characteristics of the used ion-exchange membranes in the electrolysis cell.

Membrane	IEC (mmol/g)	K_m_ (mS/cm)	W_U_ (%)	T_m_ (µm)	Permselectivity	References
Nafion^®^117	>0.9	~100	11	183	-	[35,36]
Nafion^®^911	1.1	62.5	13	213	>97	This work
Neosepta CMX	1.5–1.8	13.4	25–30	140–200	>98	[37,38]
Neosepta AMX	1.4–1.7	12.6	25–30	120–180	>98	[39,40]
Fuji AEM	1.4–1.8	9.11	24%	160	95	[41,42]
Zirfon^®^	-	37.8 in NaCl 1 M	31%	500	-	[43]
BN/PTFE	-	80.2 in KOH	33%	400	-	This work

**Table 2 membranes-12-00602-t002:** Performance and state of the used membranes after electrolysis.

Cathodic Membrane	Anodic Membrane	Chlorometric Degree (°Ch)	Membrane State
**CMX**	AMX	6.1	CMX, AMX membranes damaged
**Nafion^®^911**	BN/PTFE	1.6	Nafion^®^911 degraded after 2 tests
**Nafion^®^911**	AEM Fuji	2.87	Nafion^®^911 degraded after 2 tests
**Zirfon^®^**	Zirfon^®^	0.85	Anodic and cathodic membrane affected from the first test
**BN/PTFE**	BN/PTFE	5.8	Intact after 10 tests

## Data Availability

Not applicable.

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
