# Peer review of "A New Method Based on a Zero Gap Electrolysis Cell for Producing Bleach: Concept Validation"

_membranes, 2022, doi:10.3390/membranes12060602_

Round 1

Reviewer 1 Report

The article is written well. I recommend it to after these minor modifications.

  1. In line 117, there should be ion exchange membranes (IEMs).
  2. In line 248, there is written 99,99% instead of 99.99%. Please correct it.
  3. In line 251, there thermal properties and thermal stability. What is different between these here?
  4. In which unit, the water content was found in line 274. Mention unit here.
  5. In some locations in the manuscript, you wrote cross-section while on some location cross section. Please write in same mode.
  6. In line 419, there TGA thermogravimetric analysis. I shall suggest that TGA should be in bracket after thermogravimetric analysis. Please correct it.
  7. Explain figure 6 step by step. Please also provide DTG graph of membrane
  8. Heading No. 4 should be conclusions. Please correct it.
  9. English language required to improve throughout the manuscript.

Author Response

The article is written well. I recommend it to after these minor modifications.

  • Response: We thank the reviewer for their interest in this work as well as their constructive remarks. Below are the responses to all comments and questions.

Comment 1: In line 117, there should be ion exchange membranes (IEMs).

  • Response: Thank you for notifying us of this thoughtless error. It has been fixed in the updated version of the manuscript.

 Comment 2: In line 248, there is written 99,99% instead of 99.99%. Please correct it.

  • Response: Thank you for your attention, we have corrected this error.

Comment 3: In line 251, there thermal properties and thermal stability. What is different between these here?

  • Response: Thermal properties is a large term that includes thermal stability. We think that our sentence is badly formulated, so we suggest reformulating as follows: “Thermal properties, particularly thermal stability”.

Comment 4: In which unit, the water content was found in line 274. Mention unit here.

  • Response: the water content is the ratio of the masses which is expressed in (%).Thanks for your comment, we have put the information in the new version of the manuscript.

 Comment 5: In some locations in the manuscript, you wrote cross-section while on some location cross section. Please write in same mode.

  • Response: Thank you for your attention, we have corrected this error.

Comment 6: In line 419, there TGA thermogravimetric analysis. I shall suggest that TGA should be in bracket after thermogravimetric analysis. Please correct it.

  • Response: Thank you for your attention, you are right, we have made the necessary correction.

 Comment 7: Explain figure 6 step by step. Please also provide DTG graph of membrane

  • Response:  the TGA and DTG curves for the membrane show three main waves indicating successive degradation. Weight loss between room temperature and 120 °C is easily attributed to the residual solvent (water and ethanol) evaporation. The temperature at which polymer degradation started (Tonset) and the maximum degradation temperature (Tmax) rate of this crucial step were about 333 °C and 540 °C, respectively. Weight loss at these temperatures might be due to the degradation of the PTFE polymer starting with the short PTFE chains.

A new figure has been added in the revised version of the manuscript.

Comment 8: Heading No. 4 should be conclusions. Please correct it.

  • Response: thank you for your comment we have shortened the conclusion in the revised version of the manuscript.

Comment 9: English language required to improve throughout the manuscript.

Response: Thank you for your remarks, the manuscript was revised by a specialist in scientific English. English was checked in the new version of manuscript.

Reviewer 2 Report

The work under review provides interesting data on the membrane electrolysis for NaOCl and HOCl production. The production of said chemicals especially on low scale is of high potential commercial interest.

The proposed manuscript is well written and I recommend it to be accepted in its current form.

Author Response

The work under review provides interesting data on the membrane electrolysis for NaOCl and HOCl production. The production of said chemicals especially on low scale is of high potential commercial interest.

The proposed manuscript is well written and I recommend it to be accepted in its current form.

  • Response: We thank Reviewer # 2 for having examined this manuscript, for their appreciation of our work